# Exploiting the Features of Short Peptides to Recognize Specific Cell Surface Markers

**DOI:** 10.3390/ijms242115610

**Published:** 2023-10-26

**Authors:** Michela Buonocore, Manuela Grimaldi, Angelo Santoro, Verdiana Covelli, Carmen Marino, Enza Napolitano, Sara Novi, Mario Felice Tecce, Elena Ciaglia, Francesco Montella, Valentina Lopardo, Valeria Perugini, Matteo Santin, Anna Maria D’Ursi

**Affiliations:** 1Department of Pharmacy, University of Salerno, 84084 Fisciano, Italy or michela.buonocore@unina.it (M.B.); magrimaldi@unisa.it (M.G.); asantoro@unisa.it (A.S.); or verdiana.covelli@unina.it (V.C.); cmarino@unisa.it (C.M.); enapolitano@unisa.it (E.N.); snovi@unisa.it (S.N.); tecce@unisa.it (M.F.T.); 2Department of Chemical Sciences, University of Naples Federico II, 80138 Naples, Italy; 3Scuola di Specializzazione in Farmacia Ospedaliera, University of Salerno, 84084 Fisciano, Italy; 4Department of Pharmacy, School of Medicine and Surgery, University of Naples Federico II, 80138 Naples, Italy; 5PhD Program in Drug Discovery and Development, Department of Pharmacy, University of Salerno, 84084 Fisciano, Italy; 6Department of Medicine, Surgery and Dentistry “Scuola Medica Salernitana”, University of Salerno, 84081 Baronissi, Italy; eciaglia@unisa.it (E.C.); fmontella@unisa.it (F.M.); vlopardo@unisa.it (V.L.); 7Centre for Regenerative Medicine and Devices, School of Applied Sciences, University of Brighton, Brighton BN2 4AT, UK; v.perugini@brighton.ac.uk (V.P.); m.santin@brighton.ac.uk (M.S.)

**Keywords:** peptides, mesenchymal stromal cells, molecular docking, molecular dynamics, FACS, confocal microscopy

## Abstract

Antibodies are the macromolecules of choice to ensure specific recognition of biomarkers in biological assays. However, they present a range of shortfalls including a relatively high production cost and limited tissue penetration. Peptides are relatively small molecules able to reproduce sequences of highly specific paratopes and, although they have less biospecificity than antibodies, they offer advantages like ease of synthesis, modifications of their amino acid sequences and tagging with fluorophores and other molecules required for detection. This work presents a strategy to design peptide sequences able to recognize the CD44 hyaluronic acid receptor present in the plasmalemma of a range of cells including human bone marrow stromal mesenchymal cells. The protocol of identification of the optimal amino acid sequence was based on the combination of rational design and in silico methodologies. This protocol led to the identification of two peptide sequences which were synthesized and tested on human bone marrow mesenchymal stromal cells (hBM-MSCs) for their ability to ensure specific binding to the CD44 receptor. Of the two peptides, one binds CD44 with sensitivity and selectivity, thus proving its potential to be used as a suitable alternative to this antibody in conventional immunostaining. In the context of regenerative medicine, the availability of this peptide could be harnessed to functionalize tissue engineering scaffolds to anchor stem cells as well as to be integrated into systems such as cell sorters to efficiently isolate MSCs from biological samples including various cell subpopulations. The data here reported can represent a model for developing peptide sequences able to recognize hBM-MSCs and other types of cells and for their integration in a range of biomedical applications.

## 1. Introduction

Since the first recombinant immunoglobulin (Ig) was synthetically developed in 1984 [1,2], modern science relies on antibodies for the detection of biomolecules, given their sensitivity for specific epitopes; this allows them to be largely used as reagents of biological assays that range from enzyme-linked immunoassays (ELISA) to Western blots as well as from flow cytometry and cell sorting to immunoprecipitation. However, it is widely recognized that the use of antibodies brings various disadvantages, like a high production cost, limited tissue penetration, and potential immunogenicity, when used in in vivo diagnostic systems [3,4]. These limitations prompt the search for alternative non-Ig ligands of comparable biospecificity but with structures of higher stability and lower molecular weight, among which affibodies, single-chain variable fragments (scFv), and nanobodies have so far been proposed [5,6,7,8,9,10]. The applications of these antibody-mimetics are potentially limitless, especially in the field of immunotherapy [11]. However, the potential immunogenicity induced by engineered molecular scaffolds, especially for those derived from non-human proteins, and the low tissue penetration due to their size are still potential limitations that need to be considered [12]. The demand for miniaturizing antibody-like affinity reagents led to the successful development of scaffold-free cyclic and multicyclic peptides [4], and many works report the extraction of sequences from the paratopes of tight-binding macromolecule complexes able to recognize the corresponding epitopes [13,14,15,16]. Capitalizing on these technological progresses, it is possible to develop short peptide sequences with similar characteristics and improved pharmacokinetics.

This novel approach can lead to the development of disruptive technologies in biomedical fields where antibodies are the mainstream option.

Among them, regenerative medicine is a promising emerging field for treating patients living with chronic illnesses or with severe injuries [17,18,19]. Stem cells represent one of the most valuable instruments in regenerative medicine [20]. They have an unlimited potential for cell division and the ability to differentiate into a range of tissue cell phenotypes. In recent years, they have become an attractive therapeutic tool for repairing tissues damaged by traumas, diseases, congenital deficiencies, and age-related clinical conditions [17].

Currently, six stem cell classes are primarily involved in regenerative medicine: embryonic stem cells (ESCs), tissue-specific progenitor stem cells (TSPSCs), mesenchymal stromal cells (MSCs), umbilical cord stem cells (UCSCs), bone marrow stromal cells (BMSCs), and induced pluripotent stem cells (iPSCs) [18]. MSCs are multipotent stem cells with great regenerating potential [20,21,22]. They are situated in various tissues, like fetal, adipose, and dental; however, an efficient population of MSCs is found in the bone marrow. Bone marrow-derived MSCs (BM-MSCs) mainly develop in adipocytes, chondroblasts, and osteoblasts but also showed the property to successfully transdifferentiate in neural, myocyte, and epidermal cells when engrafted in endogenous tissues under specific conditions [23,24].

Bone marrow contains, in addition to MSCs and differentiated lineages, hematopoietic stromal cells (HSCs), which may be used less efficiently for regenerative medicine since they do not transdifferentiate when placed in other tissues [25]. Therefore, isolating MSCs from HSCs and other cell components in the bone marrow culture is strategic for successful applications of MSCs in clinic [26].

According to the International Society for Cellular Therapy (ISCT), there are three criteria to define MSCs: they must show adherence to plastic in standard culture conditions; they have to express specific surface markers (≥95%), the most interrogated ones being CD105, CD73, CD90, and CD44 and at the same time must lack expression (≥2%) of CD45, CD34, CD14 or CD11b, CD79α or CD19 and HLA-DR; they must be able to differentiate in osteoblasts, adipocytes, and chondroblasts under standard in vitro differentiating conditions [27]. Currently, the most widespread purification methods involve the isolation of MSCs based on the expression of specific surface markers. One of the most accredited protocols is Pittenger’s, which uses centrifugation by Ficoll gradient as a first step to eliminate unwanted cell types present in the marrow aspirate, then plastic adherence to remove non-adherent cells, and, finally, antibodies against markers to enhance population homogeneity [28,29,30].

In this work, we aimed to demonstrate the potential of an aptamer-based, faster, and more affordable method for the phenotypic characterization and detection of primary bone marrow-derived MSCs, the first limiting step in subsequent isolation protocols. To this purpose, we developed a strategy to identify peptide sequences to be used as an alternative to antibodies and selective towards specific MSC surface markers.

First, we selected the marker considering as criterium its expression on primary bone marrow-derived MSCs. Based on the availability of the structural and biological data [30,31,32,33,34,35,36,37], the study focused on the targeting of CD44, a receptor largely expressed on MSC surface [38,39,40]. In particular, we decided to focus on the CD44 hyaluronan binding domain (HBD) since this is the best-described moiety in the protein (UniProt ID: P16070). In fact, it is possible to see that in the AlphaFold predictions (AF-P16070-F1) [41], this is the only domain with a high Confidence Score (pLDDT > 90), while the cytosolic moieties are instead unfolded. Moreover, since it is an extracellular domain, it is a preferable target for the peptides, which are designed to recognize the surfaces of stem cells. Most importantly, the availability of several crystal structures of this moiety in complex with the natural ligand, the hyaluronic acid (HA) allowed the identification of the crucial residues involved in the receptor binding pocket. Therefore, CD44 HBD structures in complex with the HA present on the Protein Data Bank (PDB) database were used as starting points for computer-aided design of CD44 binding peptides.

To preliminary assess the robustness of the methodology and the computational data we synthesized the best resulting peptides and performed biological assays on BM-MSCs using anti-CD44 antibody as reference.

## 2. Results

### 2.1. Design of CD44 Targeting Peptides

The CD44 extracellular moiety is a receptor for HA, which is a linear glycosaminoglycan (GAG) composed of two monosaccharides: (β, 1-3)-*N*-acetyl glucosamine (GlcNAc) and (β, 1-4)-glucuronic acid (GlcUA). HA polymers are physiologically present in the organism in high (≥1 MDa) and low molecular weight (<500 kDa). While the firsts have a role in tissue regeneration and repair, the latter are released as degradation products in inflammation and induce a response by interacting with HA receptors like CD44 [42]. We considered the crystal structures of CD44 HBD (residues 23-174) in complex with HA 8-mer (PDB ID: 2JCQ, 2JCR [43]) and 4-mer (PDB ID: 4MRD [44]).

Because the crystallographic structure of the extracellular domain of human CD44 in complex with its ligand is not available in the PDB database, we decided to use the CD44 structure derived from *Mus musculus*. This is available in complex with HA, reports 87.33% sequence identity with human CD44 HBD, and includes conserved residues in the HA binding pocket (Figure 1).

Using *Mus musculus* CD44 HBD as a target (PDB ID: 4MRD [44]), we performed a molecular docking simulation (Glide) [45], defining a grid box centered on the residues highlighted in Figure 1.

Figure 2 shows the docking strategy for selecting the peptide sequences targeting the CD44 HBD. A small set of 250 peptides were screened by molecular docking simulations. The low number of structures in the starting set was due to limitations in the computational resources. The length of the peptide sequences was set of five amino acids to minimize errors due to incorrect sampling of the torsional angles in the backbone and lateral chains and perform a relatable simulation starting from limited computational resources. Moreover, the design of pentapeptides allows a rapid and inexpensive solid phase peptide synthesis (SPPS) in view of developing a low-cost protocol for isolating cell bodies.

However, the chemical properties of the natural ligand HA were considered to select the amino acids in the peptide composition: since HA includes linear β-glycosidic bonds and numerous H-bonds acceptors and donors, sequences containing aromatic and charged residues were preferred. The first run of molecular docking was carried out using the scoring function SP-Peptide, a Glide routine for the fast screening of a high number of peptides. As a result, using the docking score values as criterium (kJ/mol), seven peptides were selected. In a subsequent step, the resulting seven sequences were screened as candidate ligands using the higher accurate Glide XP scoring function and the 4-mer HA as reference compound. By analyzing the docking score values and the similarity with the binding pose of HA, we selected DTYCF and PNHSE as peptide sequences to be further tested.

Two sets of crystal coordinates are available for the human CD44 HBD receptor: an HA-bound open conformation (PDB ID: 2I83 [46]) and an HA-unbound closed state (PDB ID: 1POZ [47]); however, it was decided not to use them beforehand to design the peptides since these coordinates do not include the 3D pose of HA. Hereafter, the DTYCF, PNHSE, and reference ligand HA were redocked against these two conformations to verify if the affinity seen in the murine CD44 HBD binding pocket was reproducible in the human CD44 HBD. Analysis of the results showed that the two peptides bind both the HA-bound open and the HA-unbound closed states of the receptor with significant values of docking scores. In particular, the selected peptides interact slightly better with the receptor in the closed state thanks to their ability to reach a hydrophobic site adjacent to HBD (Figure 3). Peptides recognizing CD44 HBD in both its conformations exhibit a greater number of binding points and therefore offer additional advantages compared to the HA reference compound that did not bind the two CD44 conformations with comparable affinity.

### 2.2. Molecular Dynamics

In the context of in silico drug design, it is good practice to prove the reliability of molecular docking results using a technique that reflects a dynamic environment. Accordingly, molecular dynamics (MD) simulations were carried out using a system filled with polarized water and a non-rigid receptor to evaluate the stability and reproducibility of interactions observed in the molecular docking binding poses. MD runs were carried out using GROMACS 2020.3 [48] and CHARMM36 force field [49]. The initial receptor/ligand complexes characterized by the best interaction score were subjected to minimization and two short runs under NVT and NpT conditions and finally let run for 50 ns. Table 1 reports the main interactions established by DTYCF and PNHSE during the run compared to those seen in the docking simulations. Figure 4A and Figure 5A show the trajectory frames saved every 5 ns of the two peptides in the binding site of CD44 HBD; as demonstrated by root-mean-square deviation (RMSD) values (Figure 4F and Figure 5E), the complexes reach conformational stability before 10 ns.

MD results show that DTYCF steadily interacts with CD44 HBD throughout the whole 50 ns simulation, and its ^1^D residue establishes permanent H-bonds and salt bridges with ^45^R in the binding pocket; frequent are also the H-bonds between the ^3^Y and ^5^F of the peptide ligand and the ^46^Y and ^133^T of the receptor (Table 1). On the other hand, PNHSE binding with CD44 HBD is not equally stable: as it is possible to notice from Figure 5A, in several steps, the peptide slips out of the binding pocket, and the complex dissociates. Nevertheless, when the two entities interact, the short-range energies are rather low (Figure 5F), thanks to the high number of H-bonds established, considering that this effect might be due to the conformational sampling deriving from the docking pose.

### 2.3. Cell Assays

For the biological cell assays, the two designed sequences (DTYCF and PNHSE) were synthesized using the standard protocol of Merrifield’s SPPS [50], with and without a fluorophore tag 7-nitrobenz-2-oxa-1,3-diazol-4-yl (NBD). This molecule is excited at the wavelength of about 463 nm and emits at a maximum of 536 nm, giving a green fluorescence; NBD was added as an amino acid residue bound to D-aminopropionic acid (DAP) as the first residue in the peptide sequence. Two preliminary cell assays were performed: Western blot and the viability assay. Western blot indicates that CD44 receptor is correctly expressed on ATCC^®^ PCS-500-012 purchased hBM-MSCs (Appendix A); the viability assay proved that the vitality of the cells after the treatment with the designed peptides was not affected 24 h after the treatment and even at the highest concentrations (Appendix A). 

#### 2.3.1. Immunostaining Procedure and FACS Analysis

Fluorescence-Activated Cell Sorting (FACS) analysis was performed to verify if NBD-tagged DTYCF and PNHSE interact with the CD44 molecule on the cell surface. The fluorescence signal derived from the putative surface binding of both peptides was compared to that of a FITC-conjugated reference antibody anti-hCD44. To investigate the specificity of action and the versatility of the selected peptides, single-cell suspensions of hBM-MSC, CD44^bright^ (>96%), and Megakaryoblastic cells (MEG-01), CD44^dim^ (<57%) were used for the immunostaining procedure. MEG-01 and hBM-MSC were stained with FITC conjugated anti-human CD44 at the recommended concentration of 1 µg/test, NBD-conjugated DTYCF, and PNHSE at 2.5 µg/test, equivalent to a concentration of about 27.8 µM for DTYCF and 30 µM for PNHSE, and 25 µg/test, equivalent to 278 µM for DTYCF and to 300 µM for PNHSE.

The data shown in Figure 6 indicate that the incubation of MEG-01 CD44^dim^ cells with PNHSE and, to a greater extent, with DTYCF induced a significant increase in the fluorescent cells in terms both of percentage of positivity and Mean Fluorescence Intensity (MFI) compared to the negative control cells incubated with unstained peptides. Remarkably, both indices (percentage of positivity and MFI) were magnified when immunostaining procedures were conducted in the hBM-MSC CD44^bright^ cell population. Indeed, the incubation with increasing doses of PNHSE and, to a greater extent, with DTYCF was able to detect a higher percentage of fluorescent cells compared to the negative control cells (Figure 7).

Overall, even though the fluorescence signal was more evident in the cells immunostained with FITC-conjugated anti-CD44 reference Ab, the dose–response increase in fluorescence proportional to the expression of CD44 marker might suggest that both DTYCF and PNHSE peptides are able to bind the surface receptor CD44.

#### 2.3.2. Immunostaining

The FACS results were complemented by an immunostaining analysis enabling the study of the localization of the DTYCF and PNHSE binding areas on MSC surfaces.

MSCs were initially stained with the cell nucleus staining 4′,6-diamidino-2-phenylindole (DAPI) for a correct visualization.

Anti-CD44 Ab (0.1 mg/mL, equivalent to about 10 µM) for reference and CD44 targeting peptides DTYCF and PNHSE (0.5 mg/mL, equivalent to 558 µM for DTYCF and 601 µM for PNHSE) were added to cells at 1:50 and 1:100 dilution rates (see Section 4). Specifically, to visualize anti-CD44 and DTYCF and PNHSE peptides Alexa Fluor594 and NBD tags were used, respectively. Figure 8A shows the binding of anti-CD44 to the MSC perinuclear region as a positive control. Figure 8B,C show the results of MSC treatments with DTYCF and PNHSE: while DTYCF evidences MSCs binding at 1:100 and 1:50 dilution, no binding is evident for PNHSE at any of the concentrations used for the experiments. These data support a more significant interaction of DTYCF with MSCs compared to PNHSE, confirming the results we previously collected in the MD simulations and the FACS experiments.

Although the data shown in Figure 8 prove the binding of DTYCF in the perinuclear region of MSCs, they do not provide certainties regarding their specificity vs. the CD44 target. Therefore, we conducted a competitive assay to test the binding specificity by treating MSCs with anti-CD44 and DTYCF mixture in a 25:75 ratio. As a result, confirming the specificity of DTYCF binding vs. CD44, we observed a colocalization of DTYCF with anti-CD44 antibody in the perinuclear area of MSCs (Figure 9).

## 3. Discussion

Peptides have been advocated as a valuable alternative to antibodies in protocols of cell sorting or characterization of surface markers thanks to their easy synthesis, purification, and handling. Innovative uses of peptides have been widely explored with positive feedback from the scientific community: peptides can be efficiently used as low-weight alternatives to antibodies [9,10] or protein vaccines [51]. Moreover, because of their physicochemical properties, they have been lately rediscovered as tools to functionalize and enhance the biochemical activities of polymeric biomaterials for medical implants and tissue engineering [52,53,54,55] as well as for diagnostic devices [56,57]. In this work, two short peptide sequences were developed by the combination of rational design and in silico methodologies to target the marker CD44 expressed on the surfaces of hBM-MSCs. The peptides proved to have an antibody-comparable affinity for the mentioned marker in preliminary assays. The target CD44 has been selected considering the high expression of this marker on MSC surface [38,39,40] and the availability on the PDB database of its extracellular domain in complex with the ligand HA. Moreover, as CD44 is highly expressed in hBM-MSCs implicated in the repair mechanism of damaged cartilage, the developed detection method has the potential to be used for the sorting of cells in regenerative medicine application [58].

Starting from the structural features of the CD44-HA binding complex, a small set of 250 short peptides was screened. Molecular docking calculations led to the identification of DTYCF and PNHSE as the most promising CD44 peptide ligands. After the binding was validated by MD simulations in a dynamic system, DTYCF and PNHSE were chemically synthesized and experimentally tested for their ability to bind hBM-MSCs in FACS and immunostaining experiments.

FACS experiments were carried out on hBM-MSCs using as reference MEG-01 cells, where CD44 is expressed to a lesser extent. The data show an effective binding of DTYCF and PNHSE peptides to CD44 in both hBM-MSC CD44^bright^ and MEG-01 CD44^dim^ cells, as proved by a dose–response increase in fluorescence over a range of peptides’ concentration. It is worth noting that DTYCF binds CD44 proportionally to the expression of the receptor thus suggesting the specificity of the binding against the CD44 receptor. Moreover, this specificity was confirmed by immunostaining assessing the specific binding of DTYCF instead of PNHSE to the hBM-MSCs. DTYCF binds MSCs at 1:100 and 1:50 dilution rates, while there is no evidence for PNHSE binding at any of the concentrations used for the experiments. Finally, DTYCF showed colocalization with anti-CD44 antibody in the perinuclear area of MSCs, thus proving to be a valuable alternative to the correspondent antibody.

Taken together, the data show that using an ad hoc designed computational protocol based on molecular docking and MD calculation, it is possible to obtain a short peptide sequence with adequate biospecificity to study the expression of the CD44 receptor by an antibody-free assay. Replacing antibodies with short peptides can be a viable strategy in all procedures that require the use of antibodies to preserve sensitivity and specificity while avoiding high costs and complex laboratory protocols. This work paves the way towards the identification of a range of peptide aptamers targeting biomarkers relevant to existing or emerging biomedical applications such as regenerative medicine. In the latter case, the availability of these aptamers will enable the selection of specific cell populations suitable for tissue regeneration as such or in combination with nanoparticles.

## 4. Materials and Methods

### 4.1. Molecular Docking

Molecular docking calculations were carried out using Glide [59,60] included in the Maestro 12.3.013 software package [61]. The receptor structures of CD44 (PDB IDs: 4MRD [44], 2I83 [46] and 1POZ [47]) were prepared with Maestro’s Protein Preparation Wizard tool that allows adding hydrogens to the crystal and subsequently minimizing it with the OPLS3 force field [62] and to optimize the protonation state of His residues and the orientation of hydroxyl groups, Asn and Gln residues. The 3D structures of the peptides were generated using the Build command and prepared using the LigPrep tool, which generated tautomers and different protonation states using a pH value of 7.0 ± 2. Docking was performed considering the ligand atoms completely flexible and the receptor rigid. The grid was generated by applying a Van der Waals radius scale factor equal to 1.00 with a partial charge limit lower than 0.25e. The center of the box was measured for CD44 on residues interacting with HA (^45^R, ^46^Y, ^81^C, ^82^R, ^83^Y, ^92^I, ^98^N, ^99^A, ^100^I, ^101^C, ^102^A, ^103^A, ^104^N, ^105^H, ^109^Y). A preliminary docking calculation was performed using Glide’s Standard Precision-Peptide (SP-Peptide) mode and then repeated using Extra Precision (XP) mode. The poses with the most advantageous docking score (kJ/mol) are selected from the results, and the root-mean-square deviation of atomic position on the first pose is calculated using the *superimpose* tool.

### 4.2. Molecular Dynamics

The complexes to perform Molecular Dynamics (MD) simulations were sampled using the CD44 structure previously prepared and the peptide’s most advantageous poses obtained from docking calculations. MD simulations were run using GROMACS 2020.3 [48]. The topology files were generated using CHARMM36 all-atom force field [49]. The complexes were solvated in cubic boxes with the TIP4P water model. Na^+^ and Cl^−^ ions were added to neutralize the charge of the system. After a minimization using the steepest descent integrator, the system was equilibrated at 300 K for 1 ns as NVT ensemble and at 1 atm pressure using Berendsen algorithm NpT ensemble for 1 ns. The outputs were used for a MD simulation using Particle Mesh Ewald for long-range electrostatics under NpT conditions. Coordinates were saved every 100 ps. Trajectory files containing the coordinates of the receptor-ligand complex at different time steps (from 5 to 50 ns) were fitted in the box and converted in PDB coordinates by using *trjconv* tool of GROMACS 2020.3 Package. The structures were visualized with Maestro by Schrödinger [61]. Analyses of RMSD, number of bonds and energy were carried out for the MD simulations of each system using *rms*, *hbond* and *energy* tools of GROMACS and plotted using Microsoft Excel 16.78 [63].

### 4.3. Solid Phase Peptide Synthesis

The peptides were manually synthesized using Fmoc/tBu solid-phase peptide synthesis (SPPS) following standard C to N direction Merrifield strategy [50]. Fmoc-protected amino acids were coupled using 1-hydroxybenzotriazole and O-(benzotriazol-1-yl)-1,1,3,3-tetramethyluronium hexa-fuorophosphate (four-fold excess) as coupling reagents. At the solution was added, as an activator, six-fold excess of *N*,*N*-diisopropylethylamine. For peptide synthesis was used Wang resin. For the fluorescence based cell assays, the fluorescent amino acid L-diamino propionic acid Fmoc-Dap(NBD)-OH [64] was added to the peptides as last residue in the procedure and thus as first residue in the sequences. Cleavage and side chain deprotection was performed using a solution of 90% trifluoroacetic acid (TFA), 5% water and 5% triisopropylsilane (TIS) for 3 h. After the cleavage step, the functionalized resin was filtered using a cold solution of diethyl ether in order to precipitate the peptide. Raw peptides were purified by reversed-phase high performance liquid chromatography (RP-HPLC) using Phenomenex C18 column (length 25 cm, diameter 0.46 cm, 5 μm) under gradient elution at flow rate of 2 mL/min. The mobile phases consisted of 1% *v*/*v* TFA in water (solvent A) and in acetonitrile (solvent B). The following gradient were used: 0–30 min 10–90% B for DTYCF and 0–20 isocratic 20% B for PNHSE. Detection UV at λ = 215 nm. The sample purity was >98%. The following retention times were found: DTYCF (rt = 14.73 min), PNHSE (rt = 7.15 min), fluo-DTYCF (rt = 16.95 min) and fluo-PNHSE (rt = 11.35 min). Peptides were characterized on a Finningan LCQ Deca ion trap instrument equipped with an electrospray source (LCQ Deca Finnigan, San José, CA, USA). The samples were directly infused in the ESI source using a syringe pump at a flow rate of 5.0 mL/min. The data were analyzed using the Xcalibur 3.1 software. High-Resolution Mass Spectrometry (HRMS) electrospray ionization (ESI) *m*/*z* [M + H]^+^ Calcd. for DTYCF (C_29_H_36_N_5_O_10_S_1_) 647.70; Found 648.17. Calcd. for PNHSE (C_23_H_33_N_8_O_10_) 582.57; Found 583.33. Calcd. for fluo-DTYCF (C_38_H_44_N_10_O_14_S_1_) 896.89; Found 897.42. Calcd. for fluo-PNHSE (C_32_H_41_N_13_O_14_) 831.76; Found 832.98.

### 4.4. hBM-MSC and MEG-01 Cell Preparation

Bone marrow-derived human mesenchymal stem cells (ATCC^®^ PCS-500-012) were seeded at an initial density of 5000 cells per cm^2^ in 5 mL of complete growth medium (ATCC^®^ PCS-500-041) and incubated at 37 °C with 5% CO_2_. At a confluence of 80–90%, cells were detached using 0.05% solution of trypsin-EDTA and trypsin neutralizing solution (ATCC^®^ PCS-999-004) was used to stop trypsin digestion of cells. The split cells were centrifuged for 5 min at 1500 rpm for collection.

MEG-01 (ATCC^®^ CRL-2021) was grown in a humidified incubator at 37 °C, and 5% CO_2_ in RPMI-1640 (Gibco^®^, Thermo Fisher Scientific, Waltham, MA, USA) supplemented with 10% (*v*/*v*) fetal serum bovine (FBS, Gibco^®^, Thermo Fisher Scientific, Waltham, MA, USA), 1% (*v*/*v*) penicillin-streptomycin (Aurogene, Rome, Italy), 1% (*v*/*v*) MEM non-essential amino acids (MEM NEAA, Gibco^®^, Thermo Fisher Scientific, Waltham, MA, USA), 1% (*v*/*v*) sodium pyruvate (Aurogene, Rome, Italy).

### 4.5. FACS Assay

MEG-01 and hBM-MSC were stained with mAb against human CD44 FITC (338804-Biolegend) at the recommended concentration of 1 µg/test, NBD-conjugated DTYCF, and PNHSE at 2.5 µg/test and 25 µg/test. After 30 min incubation at 4 °C in the dark, cells were washed with staining buffer (PBS 2% fetal serum bovine plus 0.01% sodium azide), centrifuged at 1200 rpm for 5 min, and resuspended in the staining buffer for the FACS analysis. For each test, cells were analyzed using FACS Verse Flow Cytometer (BD Biosciences, Franklin Lakes, NJ, USA).

The statistical analysis was performed using GraphPad Prism 8 [65].

### 4.6. Immunostaining Assay

An immunostaining assay has been set and performed on hBM-MSCs in the presence of DTYCF, PNHSE and anti-CD44 (Abcam^®^ RabMab EPR18668, unconjugated) as control. The cells were plated at 4 × 10^4^/mL in a 24 well plates (*n* = 24) and grown for 24 h, then non-specific staining was fixed and blocked with albumin as per usual immunostaining SOP. Cells were stained with 4′,6-diamidino-2-phenylindole (DAPI). Four kinds of bindings were analyzed in the three rows of cells: in the first 0.1 mg/mL of anti-CD44 were added at two different dilutions (1:50, 1:100) and then Alexa Fluor594-conjugated secondary antibody was used to color the primary antibody; in the second 0.5 mg/mL of NBD-conjugated DTYCF were added at the two above-mentioned dilutions; in the third 0.5 mg/mL of NBD-conjugated PNHSE were added at the two above-mentioned dilutions; in the fourth both anti-CD44 and DTYCF were added in a proportion 25:75 for a competitive binding assay.

## Figures and Tables

**Figure 1 ijms-24-15610-f001:**
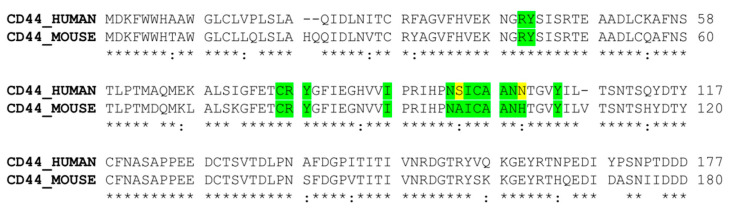
Sequence alignment between CD44 HBD deriving from human (Uniprot ID: P16070) and murine genes (Uniprot ID: P15379). The residues involved in the interaction with HA are highlighted in green when conserved and in yellow when changed in amino acids with similar properties.

**Figure 2 ijms-24-15610-f002:**
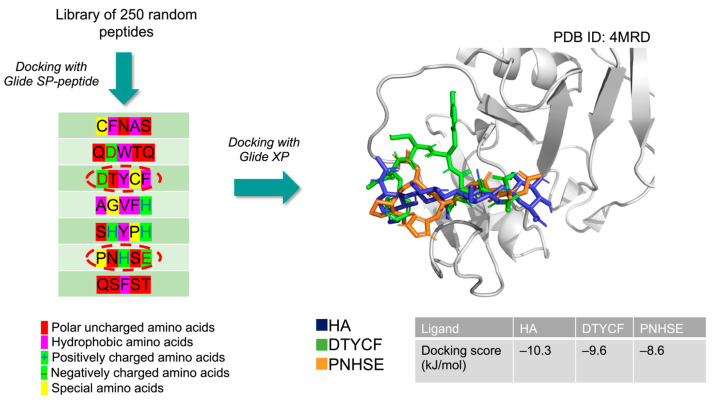
Workflow of the selection of the most promising peptides targeting murine CD44 HBD (PDB ID: 4MRD [44]): on the left are reported the sequences deriving from a preliminary molecular docking of 250 random peptides having the amino acids colored differently according to their chemical features; on the right are shown the poses of the most promising peptides (DTYCF, green sticks, and PNHSE, orange sticks) after a higher precision docking compared to HA binding pose (blue sticks) in CD44 HBD (grey ribbons).

**Figure 3 ijms-24-15610-f003:**
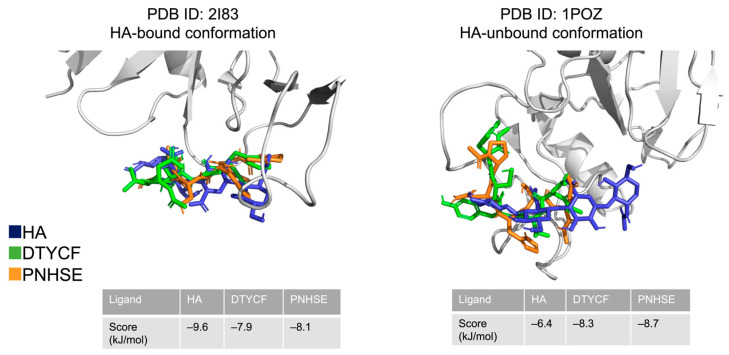
Best poses deriving from re-docking of HA (modelling images, blue sticks), DTYCF (green sticks), and PNHSE (orange sticks) with human CD44 HBD (grey ribbons) in its open HA-bound (PDB ID: 2I83 [46]) and closed HA-unbound conformations (PDB ID: 1POZ [47]). Docking score values (kJ/mol) resulted from the simulations (embedded tables).

**Figure 4 ijms-24-15610-f004:**
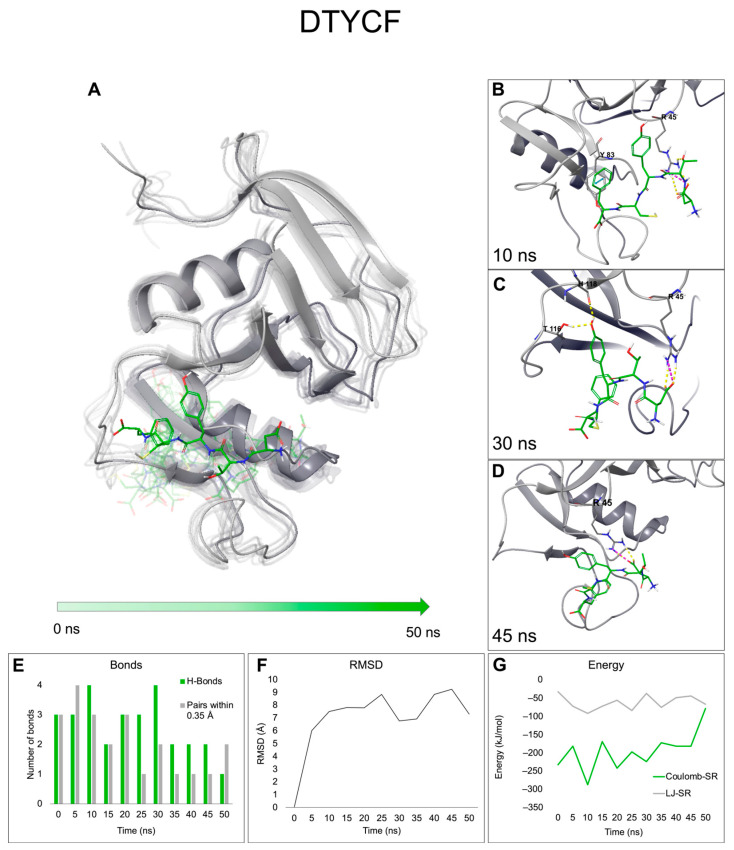
(**A**) Overlay of the frames (saved every 5 ns) of the trajectory poses obtained from the 50 ns MD simulations of CD44 (grey ribbons) in complex with DTYCF (green sticks). The complexes are here shown in increasing transparency over time. Snapshots showing the main short-range interactions between CD44 and DTYCF at 10 (**B**), 30 (**C**), and 45 (**D**) ns; H-bonds and salt-bridges are represented as yellow and pink dashed lines, respectively. Plotted representations of MD-derived data: (**E**) number of bonds (H-bonds and neighbor pairs within 0.35 nm) established between CD44 and DTYCF along the simulation, (**F**) RMSD values calculated on the protein atoms, and (**G**) Coulomb and Lennard–Jones short-range energies. All values were plotted as a function of time (ns).

**Figure 5 ijms-24-15610-f005:**
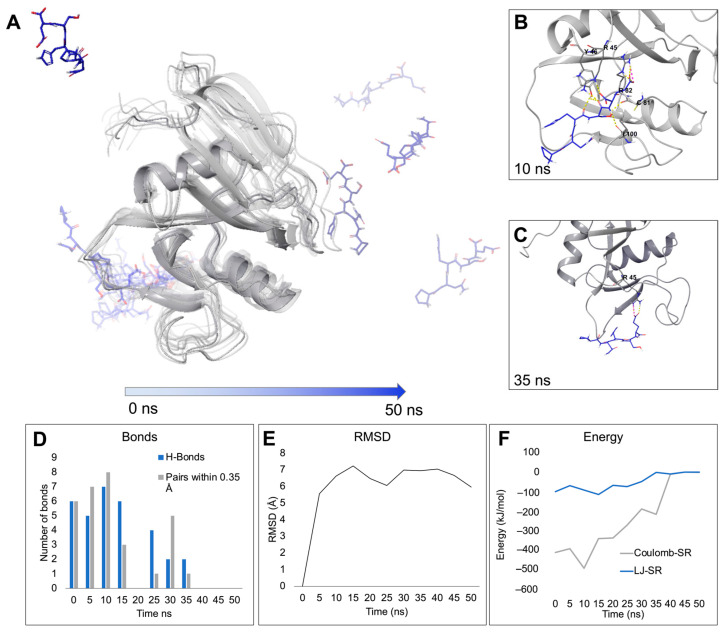
(**A**) Overlay of the frames (saved every 5 ns) of the trajectory poses obtained from the 50 ns MD simulations of CD44 (grey ribbons) in complex with PNHSE (blue sticks). The complexes are here shown in increasing transparency over time. Snapshots showing the main short-range interactions between CD44 and PNHSE at 10 (**B**), and 35 (**C**) ns; H-bonds and salt-bridges are represented as yellow and pink dashed lines, respectively. Plotted representations of MD-derived data: (**D**) number of bonds (H-bonds and neighbor pairs within 0.35 nm) established between CD44 and PNHSE along the simulation, (**E**) RMSD values calculated on the protein atoms, and (**F**) Coulomb and Lennard–Jones short-range energies. All values were plotted as a function of time (ns).

**Figure 6 ijms-24-15610-f006:**
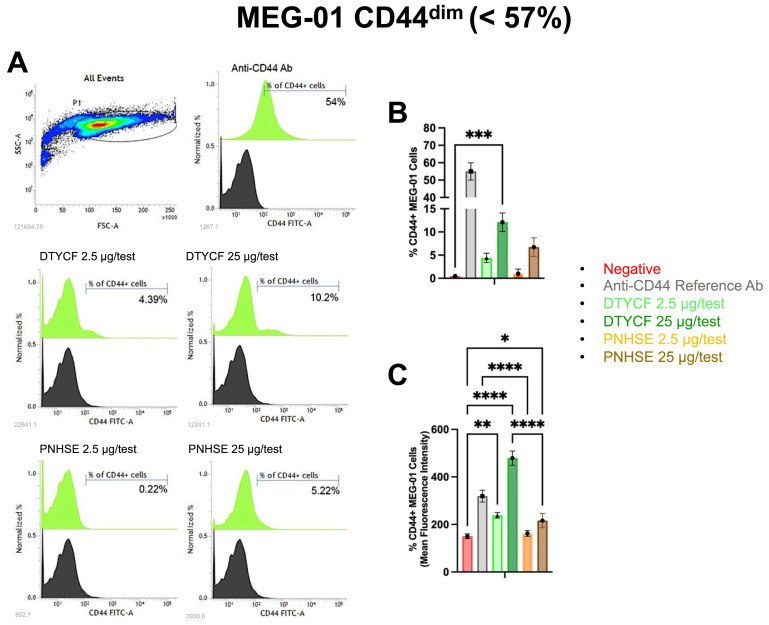
Cytofluorometric analysis of MEG-01 CD44^dim^ cells stained with NDB-DTYCF, NBD-PNHSE or FITC-Anti-CD44 antibody (Ab) as control. Panel (**A**) reports a representative FACS histogram profile of surface staining with reference Ab (**upper right**) and the two peptides (**middle** and **bottom**) at different concentrations. Black-filled curves referred to the negative control cells. Bar graphs in panels (**B**,**C**) report, respectively, the % ± SD of fluorescent positive cells and mean fluorescence intensity (MFI) values ± SD on viable gated cells. Pairwise comparisons statistically significant are indicated (ANOVA; * *p* < 0.05; ** *p* < 0.01; *** *p* < 0.001, **** *p* < 0.0001).

**Figure 7 ijms-24-15610-f007:**
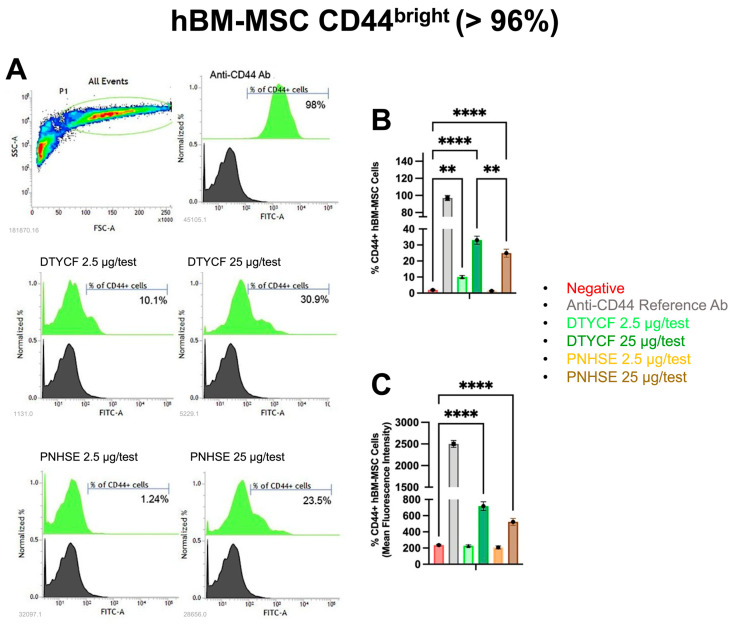
Cytofluorometric analysis of h-BM-MSC CD44^bright^ stained with NBD-DTYCF, NBD-PNHSE or FITC-Anti-CD44 Ab as control. Panel (**A**) reports a representative FACS histogram profile of surface staining with reference Ab (**upper right**) and the two peptides (**middle and bottom**) at different concentrations. Black filled curves referred to the negative control cells. Bars graph in panels (**B**,**C**) report, respectively, the % ± SD of fluorescent positive cells and mean fluorescence intensity (MFI) values ± SD on viable gated cells. Pairwise comparisons statistically significant are indicated (ANOVA; ** *p* < 0.01; **** *p* < 0.0001).

**Figure 8 ijms-24-15610-f008:**
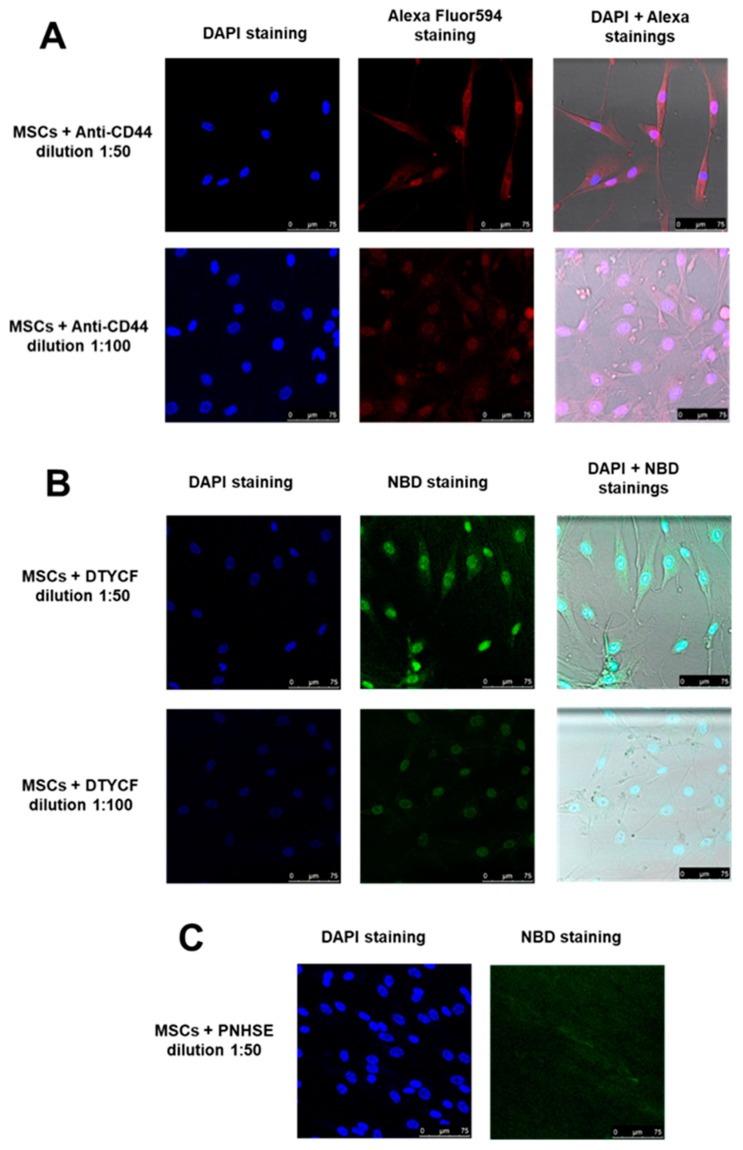
Immunostaining results: (**A**) DAPI-stained human bone marrow mesenchymal stem cells (MSCs) with Alexa Fluor594 stained anti-CD44 in two different dilution ratios (1:50, 1:100); (**B**) MSCs with NBD conjugated DTYCF in two different dilution ratios (1:50, 1:100); (**C**) MSCs with NBD conjugated PNHSE in 1:50 dilution ratio.

**Figure 9 ijms-24-15610-f009:**
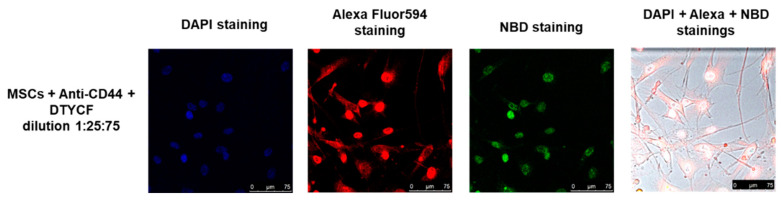
Competitive binding assay between MSCs, anti-CD44, and DTYCF in a 1:25:75 dilution ratio.

**Table 1 ijms-24-15610-t001:** Principal interactions (excluding neighbor contacts) established by DTYCF and PNHSE in the CD44 binding site as reported by molecular docking compared with the ones obtained in the 50 ns MD runs (coordinates saved every 5 ns).

	CD44 HBD-DTYCF	CD44 HBD-PNHSE
*Molecular docking*	^45^R–H-bond–^1^D ^100^I–H-bond–^1^D ^118^H–H-bond–^3^Y ^116^Y–H-bond–^3^Y	^116^T–H-bond–^2^N ^118^H–H-bond–^2^N ^83^Y–H-bond–^3^H ^46^Y–H-bonds–^3^H, ^4^S ^45^R–H-bonds–^2^N, ^4^S ^81^C–H-bond–^4^S ^100^I–H-bond–^5^E ^82^R–H-bonds–^5^E ^82^R–salt bridge–^5^E
*MD*		
5 ns	^45^R–H-bonds–^1^D ^45^R–salt bridge–^1^D ^46^Y–H-bond–^3^Y	^115^N–H-bond–^3^H ^45^R–H-bonds–^3^H, ^4^S, ^5^E ^46^Y–H-bonds–^3^H ^81^C–H-bond–^4^S ^82^R–H-bonds–^5^E ^82^R–salt bridge–^5^E
10 ns	^45^R–H-bonds–^1^D ^45^R–salt bridge–^1^D ^83^Y–H-bond–^5^F ^113^T–H-bond–^5^F	^46^Y–H-bonds–^3^H, ^4^S ^45^R–H-bonds–^4^S, ^5^E ^100^I–H-bond–^5^E ^82^R–H-bonds–^5^E ^82^R–salt bridge–^5^E
15 ns	^45^R–H-bonds–^1^D, ^2^T ^45^R–salt bridge–^1^D ^83^Y–π–π stacking–^5^F	^100^I–H-bond–^2^N ^102^A–H-bond–^2^N ^83^Y–H-bond–^2^N ^81^C–H-bond–^4^S ^46^Y–H-bond–^5^E ^45^R–H-bonds–^5^E
20 ns	^45^R–H-bonds–^1^D ^45^R–salt bridge–^1^D ^113^T–H-bond–^5^F	/
25 ns	^45^R–H-bonds–^1^D ^45^R–salt bridge–^1^D ^113^T–H-bond–^5^F	/
30 ns	^45^R–H-bonds–^1^D ^45^R–salt bridge–^1^D ^116^T–H-bond–^3^Y ^118^H–H-bond–^3^Y	^111^L–H-bond–^3^H ^83^Y—π–π stacking–^3^H ^46^Y–H-bond–^3^H ^45^R–H-bonds–^5^E ^45^R–salt bridge–^5^E
35 ns	^45^R–H-bonds–^1^D ^45^R–salt bridge–^1^D ^46^Y–π–π stacking–^3^Y	^45^R–H-bonds–^5^E ^45^R–salt bridge–^5^E
40 ns	^45^R–H-bonds–^1^D ^45^R–salt bridge–^1^D	/
45 ns	^45^R–H-bonds–^1^D ^45^R–salt bridge–^1^D	/
50 ns	^45^R–H-bonds–^1^D ^46^Y–π–π stacking–^3^Y	/

## Data Availability

Publicly available datasets were analyzed in this study. This data can be found here: https://www.rcsb.org, https://www.uniprot.org, https://alphafold.ebi.ac.uk.

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
