# Peer review of "Exploiting the Features of Short Peptides to Recognize Specific Cell Surface Markers"

_ijms, 2023, doi:10.3390/ijms242115610_

Round 1

Reviewer 1 Report

Dear all,

you will find my comments/suggestions in the attached file

Author Response

The manuscript „Exploiting the features of short peptides to recognize specific cell surface markers“, by Buonocore et al. is well written and organized, and the results are clearly presented supporting the given conclusion. The presented results look very promising offering a new possibility within the peptide field especially in the situations where the use of expensive antibodies is a requirement. Even though the sensitivity and specificity of the presented peptide sequences cannot match the ones of the corresponding antibody, the presented research opens up the opportunity some standard, very expensive and complex, procedures in future to be performed in more simple, cheaper but still efficient manner. After implementing some minor suggestions and comments that I have (please see below) I recommend acceptance in the present form:

  1. What was the main motivation to work with pentapeptides and not longer sequences?

We thank the reviewer for their constructive comments. In our experience, working with short peptides is more advantageous for in-silico calculations and synthetic strategies. When preparing the 3D structures of peptides for molecular docking, we can minimize errors due to incorrect sampling of the torsional angles in the backbone and lateral chains by designing short sequences. Thus, we can perform a relatable simulation starting from limited computational resources. Moreover, the design of pentapeptides allows a rapid and inexpensive SPPS in view of developing a low-cost protocol for isolating cell bodies.

  1. Why the authors choose to focus on the HA binding domain of CD44 and not some other

region?

We decided to focus on the HA binding domains of CD44 because this is the best-described moiety of the protein (UniProt ID: P16070), and the residues in the binding pocket are well-defined, being in complex with the HA. In fact, it is possible to see that in the AlphaFold predictions, this is the only domain with a high Confidence Score (pLDDT > 90), while the cytosolic moieties are instead unfolded. Moreover, since it is an extracellular domain, it is a preferable target for our peptides, which are designed to recognize the surfaces of stem cells. We added this information in lines 111-115.

  1. Line 393, please restructure this sentence. DIPEA is not used as scavenger, but rather as

activator in the standard SPPS protocols

We corrected the line in the manuscript.

  1. In the section “Solid phase peptide synthesis” the detected HRMS masses are 1 Da or higher

than the theoretically calculated ones. Could authors explain this? How did authors confirm that they obtained the correct peptide sequence? The difference of 1 Da or more could easily mean that during the synthesis (by mistake) another AA (with close mass to the correct one) was added. Do you have any MS/MS data that confirm your sequence’s identity?

Alerted by the reviewer's comment, we repeated the calculation of the molecular weight for the two peptides with Expasy ProtParam (Gasteiger, Elisabeth, et al. Protein identification and analysis tools on the ExPASy server. Humana press, 2005). Accordingly, we observed that the differences between the theoretical and experimental masses are below 1 Da, which can be due to the different protonation states of the molecules. We performed manual SPPS, and the HPLC runs showed a single eluted peak for both the peptides.

  1. Did the author try to selectively isolate hBM-MSCs by the described protocol?

The selective isolation of hBM-MSCs is the focus of additional ongoing work, deepening the investigation of these preliminary data in view of developing an optimized protocol.

Reviewer 2 Report

The article presents an innovative approach to using peptides instead of antibodies for targeting specific cellular markers, focusing on the example of CD44 in mesenchymal stem cells. It highlights the limitations associated with the use of antibodies, such as high production costs, limited tissue penetration, and potential immunogenicity.

The authors introduce their innovative approach of designing peptides to target CD44 on the surface of MSCs. The methodology involves molecular docking and molecular dynamics simulations, ultimately leading to the development of two peptides, DTYCF and PNHSE, with high affinity for CD44.

Experimental results are presented, demonstrating the successful binding of DTYCF and PNHSE to CD44 on hBM-MSCs. The specificity of the binding is highlighted through dose-response and immunostaining assays, indicating the potential of DTYCF as a suitable alternative to anti-CD44 antibodies.

A small set of 250 peptides was initially randomly generated for screening using molecular docking simulations, considering the HA chemical properties. However, focusing on HA chemical properties could yield many more potential candidates. Why was 250 chosen and not 2500? Are there any restrictions on the number of tests for molecular docking simulations?

Overall, this article offers a compelling solution to the limitations associated with antibodies in regenerative medicine. The method presented for designing and using peptides as substitutes for antibodies is innovative and could have far-reaching implications in the field. The combination of computational and experimental approaches enhances the credibility of the findings, making this research a valuable contribution to the scientific community.

Author Response

The article presents an innovative approach to using peptides instead of antibodies for targeting specific cellular markers, focusing on the example of CD44 in mesenchymal stem cells. It highlights the limitations associated with the use of antibodies, such as high production costs, limited tissue penetration, and potential immunogenicity.

The authors introduce their innovative approach of designing peptides to target CD44 on the surface of MSCs. The methodology involves molecular docking and molecular dynamics simulations, ultimately leading to the development of two peptides, DTYCF and PNHSE, with high affinity for CD44.

Experimental results are presented, demonstrating the successful binding of DTYCF and PNHSE to CD44 on hBM-MSCs. The specificity of the binding is highlighted through dose-response and immunostaining assays, indicating the potential of DTYCF as a suitable alternative to anti-CD44 antibodies.

A small set of 250 peptides was initially randomly generated for screening using molecular docking simulations, considering the HA chemical properties. However, focusing on HA chemical properties could yield many more potential candidates. Why was 250 chosen and not 2500? Are there any restrictions on the number of tests for molecular docking simulations?

Overall, this article offers a compelling solution to the limitations associated with antibodies in regenerative medicine. The method presented for designing and using peptides as substitutes for antibodies is innovative and could have far-reaching implications in the field. The combination of computational and experimental approaches enhances the credibility of the findings, making this research a valuable contribution to the scientific community.

We thank the reviewer for their constructive review. We started from a small set of peptides because we did not have enough resources for computational simulations and this allowed to build a combination of peptides including residues with chemical properties similar to HA (H-bond acceptors and donors, hydrophobicity). We added this consideration in lines 147-148.